# H-1 Parvovirus as a Cancer-Killing Agent: Past, Present, and Future

**DOI:** 10.3390/v11060562

**Published:** 2019-06-18

**Authors:** Clemens Bretscher, Antonio Marchini

**Affiliations:** 1Laboratory of Oncolytic Virus Immuno-Therapeutics, F011, German Cancer Research Center, Im Neuenheimer Feld 242, 69120 Heidelberg, Germany; c.bretscher@dkfz.de; 2Laboratory of Oncolytic Virus Immuno-Therapeutics, Luxembourg Institute of Health, 84 Val Fleuri, L-1526 Luxembourg, Luxembourg

**Keywords:** oncolytic virus immune therapy, rodent protoparvoviruses, H-1PV, combination therapies, second generation parvovirus treatments

## Abstract

The rat protoparvovirus H-1PV is nonpathogenic in humans, replicates preferentially in cancer cells, and has natural oncolytic and oncosuppressive activities. The virus is able to kill cancer cells by activating several cell death pathways. H-1PV-mediated cancer cell death is often immunogenic and triggers anticancer immune responses. The safety and tolerability of H-1PV treatment has been demonstrated in early clinical studies in glioma and pancreatic carcinoma patients. Virus treatment was associated with surrogate signs of efficacy including immune conversion of tumor microenvironment, effective virus distribution into the tumor bed even after systemic administration, and improved patient overall survival compared with historical control. However, monotherapeutic use of the virus was unable to eradicate tumors. Thus, further studies are needed to improve H-1PV’s anticancer profile. In this review, we describe H-1PV’s anticancer properties and discuss recent efforts to improve the efficacy of H-1PV and, thereby, the clinical outcome of H-1PV-based therapies.

## 1. Oncolytic Viruses: A General Introduction

Oncolytic viruses (OVs) are a novel class of self-propagating anticancer agents that act in a multimodal fashion to kill cancer cells [1]. The basis of their mechanism of action is the ability to selectively target, replicate in, and eventually lyse cancer cells without harming normal cells, tissues, or organs. This oncotropism can be either a natural property of the virus or the result of virus engineering at the level of virus cell entry (e.g., modification of the virus capsid to redirect the OV more specifically to receptors overexpressed in cancer cells) or virus replication (e.g., insertion of cancer-specific miRNAs into the viral promoters, which restricts replication within transformed cells).

In addition to this direct killing activity, OVs can engage the immune system in the fight against cancer [2]. Within the tumor microenvironment, a variety of different mechanisms prevent the immune system from attacking cancer cells [3,4]. OVs have the ability to reshape the tumor microenvironment and re-establish immune surveillance, thus acting as vaccine adjuvants [5]. Indeed, in addition to disseminating new progeny viral particles, OV-induced cancer cell lysis is associated with the release of danger-associated molecular patterns, pathogen-associated molecular patterns, and tumor-associated antigens, which triggers inflammatory immune responses directed against not only the virus (via the production of virus-neutralizing antibodies) but also the tumor. The immune system therefore becomes the best ally of the virus in the elimination of cancer cells, even those not directly infected by the virus (e.g., small disseminated metastasis).

Furthermore, some OVs have a natural ability to disrupt tumor vasculature, thus inducing necrosis of tumor cells due to deprivation of oxygen and nutrients [6,7,8].

Treatment of thousands of cancer patients with various OVs has demonstrated that their safety and tolerability are excellent, and that OVs are associated with only minor side effects, which are limited to flu-like symptoms such as fatigue, fever, and chills [9,10].

Talimogene laherparepvec (T-Vec or Imlygic) was the first OV approved by the US Food and Drug Administration and the European Medicines Agency, at the end of 2015, for the treatment of malignant metastatic melanoma [11,12]. It is a genetically engineered herpes simplex virus (HSV) carrying the granulocyte-macrophage colony-stimulating factor (GM-CSF), which is intended to strengthen the immune response. T-Vec reveals another interesting property of OVs: their anticancer potential can be reinforced by inserting a therapeutic transgene into their genome, for example, an apoptosis inducer with bystander effects to kill cancer cells that eventually become virus-resistant, or immune-modulators (e.g., GM-CSF) to promote more sustained antitumor immunity.

As a result of their anticancer properties, no fewer than forty OVs from at least ten families are currently being tested in clinical trials against a number of malignant indications, alone or in combination with other anticancer modalities (e.g., chemotherapy, radiotherapy, and immunotherapy) [13]. In addition to HSV, the list includes adenovirus (Ad), vaccinia virus, measles virus, coxsackie virus, poliovirus, reovirus, Newcastle disease virus, vesicular stomatitis virus, Seneca Valley virus and protoparvovirus (PV) [13]. Each of these OVs has a distinct mechanism of action, tumor tropism, immunogenicity, possibility of expressing therapeutic transgenes, potential risk of pathogenicity, stability, and specific advantages and limitations associated with the production process. These variations justify the continued development of these different virus platforms. Some of these viruses have entered late-phase clinical development and will hopefully soon become a therapeutic option for cancer patients.

Clinical studies have shown that OV treatment is often effective only in a small percentage of patients, which emphasizes the importance of developing new strategies to improve clinical outcome. As with other anticancer treatments, the combination of OVs with other therapies is believed to improve treatment efficacy. Therefore, the design of novel OV-based combination therapies is the subject of intense research for all OVs under clinical development [14]. Particularly promising are the combinations of OVs with other forms of immunotherapy (e.g., checkpoint blockade) [2,15,16,17].

In the next section, we present one of the clinically relevant OVs, the rat protoparvovirus H-1PV. We discuss its main features and clinical applications, along with recent advances in improving its anticancer activities. It is important to mention that parallel studies have been carried out also using other rodent protoparvoviruses as anticancer agents [18,19,20,21,22]. Given the focus of the review to H-1PV, these studies will not be discussed in full here.

## 2. The Rat Protoparvovirus (PV) H-1PV: A Biosketch

H-1PV was first discovered by Toolan and co-workers in the late 1950s (first publication in 1960) from transplantable human tumors [23]. It was soon realized that the infection was not causative of the tumor, but rather opportunistic, and that the virus displayed a natural tropism for human cancer cells [24]. In the 1960s and later in 1982, Toolan’s laboratory further showed that H-1PV suppressed viral- and chemical-induced tumors as well as reduced the incidence of spontaneous tumors in animal models [25,26,27]. These discoveries were seminal in establishing the concept that H-1PV’s ability to infect human tumor cells might be used therapeutically.

H-1PV is a member of the *Parvoviridae* family, genus *Protoparvovirus* (Figure 1), which also includes the Kilham rat virus, rat minute virus, LuIII virus, mouse parvovirus, minute virus of mice (MVM), and tumor virus X [28]. Some of these viruses are presently the subject of preclinical investigations aimed at evaluating their potential as anticancer therapeutics. H-1PV is among the smallest known viruses, with a diameter of 25 nm, roughly the size of a ribosome. The natural hosts of H-1PV are rats. H-1PV is shed from the animals through the feces, and transmission occurs via the oronasal route. Under normal conditions, the virus is stable for several months in the environment.

The H-1PV viral capsid contains a linear, single-stranded DNA molecule with a length of about 5100 bases. The original isolate of H-1PV was derived from an adventitious infection of the human Hep-1 hepatoma cell line, transplanted in cortisone-immunosuppressed rats [23]. Since then, the virus was further propagated in human transformed cell lines. Therefore, the current H-1PV may differ from authentic field isolates. Small differences in the length and sequence of the genome may occur naturally as a result of the virus adapting to different host cells by acquiring missense mutations or small deletions in the coding and noncoding regions of the viral genome (see below). The viral genome includes two promoters: the early P4 promoter controls the expression of the non-structural (NS) transcription unit, which encodes the nonstructural proteins NS1 and NS2; and the late P38 promoter regulates the expression of the viral particle (VP) transcription unit, which encodes the VP1 and VP2 capsid proteins and the nonstructural small alternatively translated (SAT) protein. At its extremities, the viral genome contains palindromic sequences that form hairpin structures, which serve as self-priming origins during viral DNA replication [29,30].

The 83 kDa nonstructural protein NS1 is expressed early after infection and plays multiple essential roles during the virus life cycle. NS1 activities are modulated by post-translational modifications such as phosphorylation and acetylation (see below) [31]. Owing to its ATPase and helicase activities, NS1 is the major regulator of viral DNA replication. It also plays a pivotal role in viral gene transcription, given its ability to modulate the transcription of its own P4 promoter and to activate the P38 promoter by binding specifically to DNA [32] (for a detailed review of the NS1 mechanisms of action, see Nüesch and Rommelaere, 2014 [33]). NS1 is also the major effector of virus cytotoxicity (see below), and its expression is sufficient to trigger cell cycle arrest and apoptosis—similar to expression of the whole virus [34]. The role of H-1PV NS2 is less understood, but, based on studies on the closely related parvovirus MVM, it is thought to involve the modulation of viral DNA replication, viral mRNA translation, capsid assembly, and virus cytotoxicity [35].

The H-1PV capsid, like that of other parvoviruses, consists of 60 protein subunits: 10 copies of VP1 and 50 copies of VP2 [36]. VP1 (81 kDa) and VP2 (65 kDa) are translated from the same RNA via alternative splicing, but they differ in their N-terminus. VP1 is 142 amino acids longer than VP2 (735 vs. 593 amino acids). The VP1-N-terminal region has been associated with phospholipase A2 (PLA2)-like activity and contains nuclear localization signals. Both properties are important for the transfer of the viral genome from the endocytic compartment to the cell nucleus [37,38,39,40]. In fully infectious, mature virions, but not in empty capsids lacking viral DNA, VP2 undergoes proteolytic cleavage of 18–21 amino acids at its N-terminus to form VP3, which becomes the major component of the viral capsid [36].

Structural crystallographic analysis of the H-1PV capsid has revealed the typical capsid of parvoviruses: a cylindrical structure surrounded by a canyon-like depression at the fivefold axes, spike protrusions at the icosahedral threefold axes, and a dimple-like depression at the twofold axes that appears to be involved in cell-surface recognition and binding [36].

The H-1PV-specific cellular receptor(s) remain to be identified, although terminal sialic acid has been shown to play an essential role in H-1PV cell surface binding and entry [41]. Indeed, treatment with neuraminidase, which cleaves sialic acid from the cellular surface, strongly reduces H-1PV infection by impairing virus cell attachment. The essential role of sialic acid in H-1PV cell surface recognition was confirmed in Chinese hamster ovary (CHO) cells. Whereas the parental CHO Pro-5 cells, which express sialic acid on their surface, are fully susceptible to H-1PV infection, the two isogenic CHO Lec 1 and Lec 2 mutants, which lack sialic acid, are resistant. Two residues at the twofold depression, I368 and H374, are essential for binding to sialic acid [41].

Based on homology with other members of the *Parvoviridae* family, H-1PV cell entry is believed to occur after virus cell membrane binding, via clathrin-mediated endocytosis (for a comprehensive review of PV entry mechanisms, see the review from Ros et al., 2017 [42]). However, the H-1PV cell entry pathways remain to be elucidated. After trafficking into the cytosol, H-1PV penetrates the nucleus. For its viral DNA replication, the virus needs proliferating cells but is itself unable to induce a quiescent cell to enter S-phase. Once the cell enters S-phase, the single-stranded genome is converted to the active double-stranded forms that are accessible for transcription [42,43]. As soon as empty capsids are assembled into the cell nucleus, the single-stranded viral genome is transferred to the shells, and the progeny viruses are transported to the cytoplasm. At the completion of its life cycle, the virus induces cell lysis, which is associated with the extracellular release of progeny viral particles. These new virions can initiate second rounds of lytic infection in neighbouring cells [42].

## 3. H-1PV at the Preclinical Level: Acquiring License to Kill Cancer Cells

In this paragraph we discuss the main features that make H-1PV an attractive oncolytic virus (OV).

### 3.1. Non-Pathogenicity in Humans

The natural host of the H-1PV is the rat. Humans are naturally not infected with the virus. No link has been established between the virus and human diseases, and no pre-existing immunity to H-1PV has been demonstrated in humans. The latter represents an advantage of H-1PV over OVs based on human pathogens (e.g., HSV and Ad), as H-1PV may have a larger therapeutic window before the appearance of neutralizing antibodies. Laboratory studies have demonstrated that although H-1PV can penetrate normal, non-transformed cells, this infection fails to produce new virus particles (i.e., it is an abortive infection) and to induce cell lysis [44]. Clinical studies have shown that H-1PV treatment is safe, well tolerated, and not associated with unwanted side effects (see below).

### 3.2. Natural Oncotropism

Because of H-1PV’s limited genomic information, its life cycle is strictly dependent on the host cell. Some of the factors needed for a productive virus infection are more abundant or more specifically active in the cancer cell than in its normal counterpart. Thus, the cancer cell provides a more favourable milieu than the normal cell for sustaining the virus life cycle. The determinants of H-1PV oncoselectivity have been the focus of several recent reviews and are not discussed in detail here [18,31,44]. Some of these interactions have been described for the closely related MVM and are believed to take place for other PVs, including H-1PV. Briefly, PVs take advantage of some of the (epi)genetic defects that distinguish cancer cells (listed below) at many stages of their life cycle.

#### 3.2.1. Uncontrolled Proliferation

PV DNA replication and, in particular, the conversion of the single-stranded genome into the active double-stranded form rely on cellular factors (e.g., the cyclin A/CDK2 complex) that are S-phase-specific and typically expressed in proliferating cells [45,46].

#### 3.2.2. Dysregulated Signaling Pathways

Various factors that are overexpressed in cancer cells are active in controlling PV nuclear transfer (e.g., CDK1/PKCα-mediated rupture of the nuclear envelope [47]), NS1 activities (e.g., PDK1/PKB/PKC involvement in the phosphorylation of NS1 [48]), viral gene expression (e.g., members of the E2F, Ets, and ATF families of transcription factors are needed to activate the P4 promoter [44,47,49]), virus replication (e.g., interaction with components of the DNA damage response, such as RPA-P32, γH2AX, NBS1-P, ATR, ATRIP, and ATM, which are recruited in the subnuclear PV replication centres, the so-called APAR bodies [50]), viral progeny capsid assembly and nuclear transport (e.g., MAP3K-mediated phosphorylation of capsid intermediates [51]), and virus egress (e.g., XPO1, PKB, PKCη, and Radexin, which regulate various steps involved in trafficking of the virus outside the cell [52,53,54]).

#### 3.2.3. Impairments of Innate Antiviral Immunity

Defects in the innate immune system are common in cancer cells, which often makes them unable to counteract a virus infection efficiently. H-1PV infection, similarly to MVM infection, triggers an antiviral innate immune response that is associated with the production of type I interferons (IFNs) in normal cells but not in cancer cells. This antiviral response efficiently blocked H-1PV multiplication only in normal cells [21]. However, the sensitivity of rodent PVs to type I IFNs is presently a matter of scientific discussion [20,55,56,57,58].

All these interactions (and probably many others still to be characterized) define whether a certain cancer cell is susceptible or not to H-1PV infection. The discovery of new H-1PV cellular modulators is extremely important, as these signatures may serve as markers to predict if a certain patient is likely to respond favorably or not to H-1PV treatment (see also below).

### 3.3. Oncolytic Activities

Cancer cell lines and primary cultures derived from various tumor entities, including brain, pancreas, breast, lung, cervical and colorectal cancers, melanoma, and osteosarcoma, are susceptible to H-1PV infection and oncolysis (reviewed in [18]). H-1PV was also shown to efficiently infect and kill cancer cell lines derived from hematological diseases such as Burkitt lymphoma, diffuse large B-cell lymphoma, T-cell acute lymphoblastic leukaemia, and cutaneous T-cell lymphoma [59]. Both apoptosis and non-apoptotic cell death have been reported to be induced by H-1PV [34]. Furthermore, in glioma cells, H-1PV induces lysosome-dependent cell death with relocation of active cathepsins B and L (CTSB and CTSL) from lysosomes into the cytosol and concomitant repression of two cathepsin inhibitors, cystatin B and C [60]. By inducing this alternative cell death pathway, H-1PV is able to overcome glioma cell resistance to conventional cytotoxic agents like cisplatin or to soluble death ligands such as the pro-apoptosis inducer TNF-related apoptosis-inducing ligand (TRAIL).

The reasons why H-1PV induces lysosome-dependent cell death in glioma cells, but apoptosis or other forms of cell death (e.g., necrosis) in other cancer cell lines, were recently investigated by our laboratory. We discovered that pro-survival members of the BCL2 family (e.g., BCL2, BCL2L2, BCL2L1, and MCL1), which are overexpressed in glioma (and other tumor) cells and contribute to their resistance to apoptosis inducers, acted as negative modulators of H-1PV-induced apoptosis. Indeed, the addition of BH3 mimetics such as ABT-737 (which inhibits pro-survival BCL2 proteins) rescued the ability of H-1PV to induce apoptosis in these cells, thereby strongly potentiating H-1PV glioma cell oncolysis [61]. 

H-1PV-induced cell death is mediated by NS1 through the accumulation of reactive oxygen species, which leads to oxidative stress, mitochondrial outer membrane permeabilization, DNA damage, cell cycle arrest, and, finally, caspase activation [34].

H-1PV-induced cell death is also associated with several markers of immunogenic cell death, such as release of the high-mobility group box protein B1 [62] and the immunogenic heat shock protein HSP72 [63]. In a co-culture experiment in which melanoma cells were grown together with dendritic cells (DCs), H-1PV-induced cell lysis stimulated DC maturation and activation [64], accompanied by the production of proinflammatory cytokines such as IL-6 and TNF-α. Mature DCs were able to activate antigen-specific cytotoxic T cells, which resulted in IFNγ production (discussed in the issue by Angelova and Rommelaere [58]).

### 3.4. Oncosuppressive Activities

The oncosuppressive activities of H-1PV have been demonstrated in various animal models (reviewed in [18,44]). Oncosuppression is a result not only of H-1PV tumor oncolysis, but also of the activation of immune responses. The immunostimulatory activities of H-1PV are discussed in the issue by Angelova and Rommelaere [58]. As examples of the oncosuppressive activities of H-1PV, here we summarize experiments carried out in animal models of glioma and pancreatic ductal adenocarcinoma (PDAC) [58].

#### 3.4.1. Glioma Models

In an immunocompetent rat model in which RG2 rat glioma cells were implanted into the brain of allogenic Wistar rats, intratumoral treatment with a single dose of H-1PV (1 × 10^7^ plaque forming units/animal) significantly increased the overall survival of tumor-bearing animals, with one-third of the treated animals undergoing complete tumor remission [65]. Similar anticancer activity was achieved after systemic or intranasal delivery of H-1PV, although higher concentrations of virus were required in comparison to local injection [65,66]. These experiments show the ability of H-1PV to cross the blood-brain barrier in order to reach tumor cells. The NS1 viral protein was detected in tumors but not in normal surrounding tissues, confirming the oncoselectivity of H-1PV. The expression of the oncotoxic viral protein was associated with higher levels of CTSB, confirming previous results obtained in cell culture models [60]. Furthermore, progeny viruses were isolated from the animals, providing evidence of efficient virus multiplication in tumors [65] but not in other organs or tissues [67,68]. Virus treatment was not associated with weight loss or other adverse toxic events [65,66,68], even when the virus was directly injected into the brain of naive rats at high concentrations [67].

Importantly, involvement of the immune system in the elimination of cancer cells was also demonstrated, as antibody depletion of CD8+ T cells strongly reduced virus-mediated oncosuppression [69].

The oncosuppressive activity of H-1PV was also confirmed using the U87 xenograft model of human gliomas in immunodeficient rnu rats [65]. Rnu rats lack a normal thymus and, thus, cannot form T cells. In this model, T cells seem to be dispensable for H-1PV oncosuppression. However, it is not possible to exclude that other immune components such as macrophages and natural killer (NK) cells, which are still functional in rnu rats, may have participated in the elimination of cancer cells, thus compensating for the absence of T cells. In support of this hypothesis, H-1PV was shown to stimulate NK anticancer activity [70].

#### 3.4.2. Pancreatic Ductal Adenocarcinoma (PDAC) Models

H-1PV was used alone or in combination with gemcitabine, the first-line treatment for PDAC. In a syngeneic orthotopic rat model of PDAC, H-1PV treatment alone prolonged animal overall survival. However, a stronger anticancer activity was observed when the virus was combined with gemcitabine [71]. H-1PV’s ability to replicate in PDAC cell lines positively correlated with SMAD4 expression levels. Indeed, it was seen that SMAD4 bound to the P4 promoter, thereby modulating its activity [72]. A large set of experiments confirmed in PDAC models the central role of the immune system in H-1PV-mediated oncosuppression. H-1PV has the ability to evoke both innate and adaptive immune responses, as discussed in detail by Angelova and Rommelaere [58].

## 4. H-1PV Goes to Patients: Meeting the First Endpoints

First clinical use of H-1PV for the treatment of cancer goes back to 1965 [73]. These studies provided first evidence that H-1PV treatment was safe, although, at the regimes used, it did not alter the course of the patients’ cancers. This evidence, together with the subsequent preclinical results described above, laid the foundations for the launch in 2011 of a phase I/IIa clinical trial (named ParvOryx) using H-1PV for the treatment of patients suffering from recurrent glioblastoma (GBM) (see also Angelova and Rommelaere, this issue [58]). GBM is the most aggressive and common type of primary malignant brain tumor in the adult brain. GBM remains uniformly fatal, with a dismal median overall survival of only 12–15 months and with only 4.5% of patients surviving more than 5 years. Hence, new therapeutic options are urgently needed [74]. ParvOryx was the first clinical trial in Germany to use OVs. The study involved 18 patients, subdivided into two arms that were treated with escalating doses of H-1PV administered intratumorally or intravenously. The results of the study are summarized in Figure 2. The trial met its endpoints by demonstrating that monotherapy with H-1PV is safe and generally well tolerated. H-1PV showed the ability to cross the blood–brain barrier, to distribute widely in the tumor microenvironment, and to trigger inflammatory responses, confirming previous results obtained at the preclinical level. Compared to historical controls, progression-free and overall survival of the patients was improved, although all patients ultimately died from the disease. A randomized, double-blind study needs to be performed to unequivocally demonstrate the efficacy of H-1PV treatment.

A second clinical study (ParvOryx02), launched in 2015, used H-1PV to treat patients with PDAC. PDAC is one of the most lethal forms of human cancer, with a five-year survival rate of about 6% and a median patient survival rate of less than six months after diagnosis [75]. ParvOryx02 involved a total of seven PDAC patients with at least one liver metastasis. Escalating doses of H-1PV were given intravenously (40% of the dose subdivided in four equal daily fractions) and locally into liver metastases (60% in one single treatment) [76]. Recruitment has been completed, and the study is presently in its evaluation stage. Safety and tolerability are the main endpoints of the study, while evaluation of antitumor activity and clinical efficacy are the secondary objectives.

## 5. H-1PV Back to the Bench: Further Improving Its Anticancer Profile

As discussed in the previous sections, preclinical and clinical results using wild-type H-1PV as a monotherapy are promising and support its use as an anticancer agent. Nevertheless, these results also show that, as seen for other OVs, there is a discrepancy between H-1PV anticancer efficacy achieved at the preclinical level (e.g., in animal models) and that observed in patients. The fact that H-1PV treatment did not eradicate the tumors clearly indicates the need to improve its efficacy.

The field of oncolytic virus therapy in the last years shifted from considering the OVs as self-amplifying drugs able to directly kill cancer cells by inducing their lysis, to a form of immunotherapy acting indirectly through the induction of anticancer immune responses. However, it remains unclear how many rounds of lytic cycles are needed to harness the immune system to act against the cancer. The clinical experience gathered in these years indicated that, with the exception of few anecdotic cases, treatment with OVs was unable to eliminate all cancer cells. Even in the cases where virus treatment resulted in shrinkage of the tumor and induction of anticancer immune responses, tumors were not completely cured and eventually relapsed. It is possible that the high heterogeneity of tumors and the immunosuppressive nature of the TME have helped some tumor cells to survive the treatment.

We believe that, like for other OVs H-1PV efforts should also be further directed not only to enhance the immune modulatory activities of the virus but also to increase virus multiplication, spread, and oncolysis in the tumor bed. If a larger number of cancer cells are targeted and killed by the virus in the first place, the induction of anticancer immune responses is likely to be more robust.

Also of primary importance is the identification of reliable biomarkers that could be used to identify those patients most likely to benefit from H-1PV-based anticancer treatments. Currently, several approaches are being pursued in an attempt to improve the anticancer potential of H-1PV (Figure 3). Preclinical proofs of concept for some of these approaches have already been acquired, warranting clinical translation of these novel therapies.

### 5.1. H-1PV-Based Combination Therapies

Cancer heterogeneity often limits the efficacy of a single anticancer treatment, rendering it unable to eliminate all cancer cells. A common trend in anticancer therapy is, therefore, the rational design of novel combinatorial treatments that combine two or more agents with complementary mechanisms of action, leading to additive or better synergistic anticancer effects without increasing adverse events. Similarly, a logical approach to improve the efficacy of H-1PV (and, in general, any OV) is to search for other anticancer modalities that increase virus potency while preserving the safety profile (Figure 3).

#### 5.1.1. H-1PV in Combination with Conventional Treatments

For the sake of expediting clinical translation, OVs have been combined with first-line treatments such as radiotherapy and chemotherapy, often with encouraging results [77,78]. Geletneky and colleagues showed that radiotherapy sensitized low-passage cultures of human glioma to H-1PV treatment [79]. In particular, pre-irradiation 24 h before H-1PV infection increased the fraction of glioma cells in S-phase, thereby rendering the cells more susceptible to H-1PV replication. This effect led to increased cell killing even in radiation-resistant glioma cells. However, this promising protocol has yet to be validated in animal models. As radiotherapy also has immunostimulatory activity [80], it would be interesting to verify whether co-treatment also results in more sustained anticancer immune reactions.

Gemcitabine is a chemotherapeutic drug that acts as a cytidine analogue. However, its use is often accompanied with high toxicity and limited efficacy due to quick acquisition of drug resistance by the cancer cell [81]. H-1PV/gemcitabine co-treatment showed an additive killing activity in vitro that was associated with higher levels of cathepsin B, suggesting that co-treatment triggered a lysosomal cell death pathway. Higher levels of HMGB1 danger signalling were also observed, providing some indications that the killing activity induced more sustained anticancer immune responses [62].

H-1PV efficiently infected and killed gemcitabine-resistant PDAC cells, thus circumventing the drug resistance of the cancer cells. By contrast, gemcitabine pretreatment seemed to potentiate H-1PV anticancer activity through still uncharacterized mechanisms. In a syngeneic orthotopic rat model of PDAC, the consecutive combination of gemcitabine and H-1PV increased the overall survival of tumor-bearing animals with no apparent unwanted cytotoxic effects [71], warranting clinical translation of such a protocol. Further studies are required to optimize these protocols and find the most opportune specific regimes, sequence of addition, and temporal schedule of treatment.

#### 5.1.2. H-1PV in Combination with Epigenetic Modulators

The histone deacetylase inhibitor (HDI), valproic acid (VPA), significantly increases the oncolytic activity of H-1PV. VPA and other HDIs have been shown to induce cell cycle arrest and apoptosis in cancer cells [82]. VPA, which is currently in clinical use for the treatment of epilepsy, is being tested in various clinical trials as an anticancer agent, alone or in combination with other drugs (source: https://clinicaltrials.gov). At doses within the clinical range used for long-term treatment of epileptic patients and sublethal for cancer cells, VPA boosted the oncolytic activity of H-1PV in a synergistic manner in cancer cell lines derived from cervical and pancreatic carcinomas but not in normal, non-transformed cell cultures [83]. Synergistic anticancer activity was attributed to the ability of VPA to increase the acetylation status of the NS1 protein. Two lysine residues, K85 and K257, were found by mass spectrometry to be acetylated within NS1, and their acetylation levels were enhanced by the addition of VPA. Acetylated NS1 has increased DNA binding and transcriptional activities, which resulted in enhanced virus replication in tumor cells. Co-treatment with H-1PV and VPA was associated with a significant increase in oxidative stress associated with accumulation of intracellular reactive oxygen species and DNA damage. Enhanced H-1PV replication and oxidative stress contributed to the synergistic killing activity. However, because of the pleiotropic action of VPA, the role of other uncharacterized mechanisms (e.g., VPA-mediated modulation of the innate immune response) could not be excluded. Validation of the protocol in animal models showed that VPA strongly enhanced the oncosuppression activity of H-1PV, which resulted in complete and long-lasting tumor remission in all co-treated animals under conditions in which single treatment had no, or only slight, benefits for animal survival. This outcome was accompanied by higher virus multiplication, oxidative stress, and DNA damage, thus confirming the results obtained in cell culture experiments [83].

#### 5.1.3. H-1PV in Combination with Apoptosis Inducers

More recently, the BH3 mimetic ABT-737 has also been found to act synergistically with H-1PV [61]. ABT-737 is an inhibitor of pro-survival/anti-apoptotic Bcl-2 proteins, which are involved in the regulation of apoptosis. Defects in the apoptotic pathways occur frequently in cancer cells. One of the most common mechanisms by which cancer cells counteract apoptotic stimuli is the overexpression of Bcl-2 proteins [84]. The addition of the drug significantly potentiated H-1PV oncolysis against a large panel of cancer cell lines derived from solid tumors including gliomas, pancreatic carcinomas, and cervical carcinomas as well as lung, head and neck, breast, and colon cancers. This strong anticancer effect was also observed in cancer cell lines that were poorly susceptible to H-1PV oncolysis. More recently, the co-treatment was validated in xenograft rat models of human glioma and pancreatic carcinoma. In these animal models, ABT-737 significantly boosted H-1PV-mediated oncosuppression, resulting in a significant increase in animal overall survival (unpublished results). Further studies are required to determine the mechanisms underlying the synergistic anticancer effect and to verify whether increased oncolysis leads to more robust stimulation of anticancer immune responses.

#### 5.1.4. H-1PV in Combination with Antiangiogenic and Immune-Modulating Drugs

Because of their immunostimulatory activities and ability to convert an immunosuppressive cold tumor microenvironment (TME) into an inflamed one, OVs are presently under evaluation as boosters of other forms of immunotherapy with very promising results obtained at both preclinical and clinical levels [2,12,85]. The combination of H-1PV with other immune therapeutics (e.g., checkpoint blockade) holds great promise, as discussed in the issue by Angelova and Rommelaere [58] and in other recent reviews [69,86]. This promise was exemplified by a recent report describing nine patients with primary or recurrent glioblastoma who were treated as part of a compassionate use program with a combination of H-1PV, the antiangiogenic antibody bevacizumab, and the PD-1 checkpoint blockade, nivolumab [87]. This study strongly supports the combination of H-1PV with antiangiogenic drugs and checkpoint blockade and warrants further investigation to define optimal treatment regimes. Unfortunately, preclinical evaluation of combined treatments involving H-1PV and immune checkpoint blockade (e.g., with antibodies against PD-1, PD-L1) is hampered by the lack of a mouse cancer cell line that is permissive to H-1PV infection, which precludes the use of syngeneic mouse tumor models. However, as the cancer immunotherapy field is progressing rapidly, antibodies against rat immune-checkpoint proteins may become available in the near future, rendering possible the use of rat models that are susceptible to H-1PV treatment.

### 5.2. Second-Generation Propagation-Competent H-1PV-Based Vectors

The strategies pursued thus far to improve the anticancer properties of H-1PV are listed below and summarized in Figure 3B.

#### 5.2.1. H-1PV Fitness Mutants

H-1PV is a fast-evolving virus that can adapt to a specific host cell environment by acquiring spontaneous genetic modifications. A naturally occurring H-1PV variant was isolated in a newborn human kidney cell line NB-E in the course of routine plaque purification [88]. The virus featured a 114 nucleotide (nt) in-frame deletion (nt 2022–2135 of the viral genome) encompassing the NS region and a duplication of a 58 nt repeated sequence within the right-hand palindrome. As a consequence of the deletion, NS1 and NS2 proteins lost 38 amino acids at the C-terminus and internally, respectively. The deletion conferred to the virus a superior fitness at the level of nuclear export and spreading compared to wild-type H-1PV [89]. In a subsequent study, Hashemi et al. explored the effects of mutations within the H-1PV NS encoding region. By introducing into the H-1PV genome single nucleotide changes that have been shown to improve the fitness of the closely related lymphotropic strain of MVM, the authors generated H-1PV fitness variants with enhanced infectivity and transduction efficiency [35].

In another study, Nuesch et al. generated a number of adapted variants by serially passaging H-1PV in semipermissive, low-passage human glioma cell cultures. The variants contained small deletions and/or point mutations leading to single amino acid substitutions within both the coding (NS and VP gene units) and the untranslated regions of the viral genome [90]. Similar to previous studies, small deletions were found between nts 2000 and 2200 of the viral genome, suggesting that this part of the genome may represent a hotspot of variability for adapting the virus to a certain cell host. The adapted viruses displayed greater capacity to replicate in glioma cells and increased infectivity. The evaluation of the oncolytic activity of these fitness mutants, both in cell culture and animal models, together with the assessment of their safety profile is an interesting area of future research.

#### 5.2.2. H-1PVs Armed with Immune Stimulators

The limited packaging capacity of PVs allows the insertion of only small heterologous DNA sequences (max. 250 bases) into their genome (the insertion of larger transgenes can only be at the expense of the VP region, rendering the recombinant PVs replication deficient [91]).

Raykov et al. inserted CpG motifs into the untranslated region of the H-1PV genome downstream of the VP gene unit [92]. These sequences are frequently found in the genomes of microbes and have immunostimulatory activities. Insertion of CpG elements into the virus genome did not affect virus replication and infectivity. CpG-armed viruses were endowed with enhanced immunogenicity and adjuvant capacity in both cell culture and animal models [58,92,93].

#### 5.2.3. H-1PVs Armed with RNA Interference Triggers

Tumors are often highly heterogeneous in nature. Within a certain tumor, a fraction of cells may be moderately susceptible to H-1PV infection and survive virus treatment, leading to tumor relapse. RNA interference technology is used to silence the expression of genes involved in carcinogenesis in order to revert the malignant phenotype. To potentiate H-1PV oncotoxicity and provide the virus with an additional mode of action for killing those cancer cells that are poorly sensitive to its infection, we inserted single hairpin RNA (shRNA) expression cassettes into the untranslated region of the H-1PV genome. In a proof of concept study, we showed that the new virus, which we called H-1PV silencer, was able to express shRNAs at high levels and was efficient in gene silencing while retaining its ability to replicate and propagate efficiently [94]. More recently, we constructed an H-1PV silencer expressing shRNAs targeting CDK9 (H-1PV sil-shCDK9), whose expression and activity are often dysregulated in cancer cells, thus contributing to cancer development. H-1PV sil-shCDK9 has superior oncolytic activity in semipermissive pancreatic- and prostate-derived cancer cell lines in comparison with wild-type virus. Validation of these results in xenograft nude rat models of human pancreatic (AsPC-1) and prostate (PC3) carcinomas confirmed the stronger anticancer activity of H-1PV sil-shCDK9, which led to a significant increase in the overall survival of treated animals. These results warrant further development of this promising approach.

#### 5.2.4. Cancer Retargeted H-1PVs

Although H-1PV preferentially expresses and replicates its genome in (pre)neoplastic cells, it is also able to infect normal cells in a nonproductive way in which it is harmless for the cells. However, uptake of the virus by normal cells sequesters a significant portion of the administered viral dose away from the tumor target, thus reducing its efficacy [95]. It would be beneficial to limit H-1PV entry specifically to cancer cells, especially in view of the systemic delivery of the virus in therapeutic applications. Allaume et al. showed that it is possible to genetically engineer the H-1PV capsid and modify the tropism of the virus at the level of virus entry [41]. Based on an in silico model (Figure 1), the authors identified two putative residues involved in the binding to sialic acid at the twofold axis of symmetry of the virus capsid. Amino acid exchange at one of these sites (H174R) strongly reduced cell surface binding and entry without affecting virus capsid formation. This mutant was used as a template for the insertion of an arginine-glycine-aspartic acid (RGD)-4 cyclic peptide, known to bind α_V_β_3_ and α_V_β_5_, two integrins that are often overexpressed in cancer cells and angiogenic blood vessels [96]. Insertion of the peptide in one of the most protruding loops of the threefold spike of the virus capsid rescued virus infectivity and conferred to the virus improved specificity for cancer cells.

#### 5.2.5. Adenovirus (Ad)–PV Chimera

To combine the high titre and efficient gene transfer capacity of Ad with the anticancer potential of H-1PV (PV), an engineered version of the H-1PV genome was inserted into a replication-defective (E1- and E3-deleted) Ad5 vector genome to create an Ad–PV chimera [97]. The Ad carrier serves as a Trojan horse to bring the H-1PV genome into cancer cells, where the PV DNA is excised from the Ad backbone and autonomously initiates a genuine PV cycle, resulting in the production of PV particles. These PV particles retain the ability to infect neighbouring cancer cells, kill them, and induce secondary rounds of lytic infection, thereby amplifying the initial cytotoxic activity of the chimera (Figure 4). As a consequence, the Ad–PV chimera exerts stronger cytotoxic activities against various cancer cell lines than those of the PV and Ad parental viruses while still being innocuous to a panel of normal primary human cells. The Ad–PV chimera also offers the advantage of overcoming the limited cargo capacity of the PV. Indeed, the Ad backbone can accommodate therapeutic transgene(s) encoding pro-apoptotic or immunostimulating factors, whose activity may reinforce the anticancer effect of PV. Therefore, the chimera offers the advantage of combining in only one vector, effectors for both cancer gene therapy (non-propagating Ad-mediated delivery and expression of therapeutic transgenes in cancer cells) and oncolytic viro-immunotherapy (PV particles retaining oncolytic and anticancer adjuvant properties as well as the capacity for propagating in the tumor bed).

### 5.3. Recombinant Propagation-Deficient H-1PV-Based Vectors

As reported above, protoparvoviruses have a limited packaging capacity. As a result, only small heterologous DNA sequences can be inserted in the parvoviral genome, without impairing the ability of the virus to self-propagate. Arming protoparvoviruses with larger transgenes is still possible by replacing a part of the VP gene unit with a therapeutic gene [95,98]. These recombinant PVs (recPVs) retain the NS1/2 coding sequences (controlled by the parvoviral P4 promoter) and the parvoviral genome telomeres, which are necessary for viral DNA amplification and packaging. Expression of the transgene is generally kept under control of the genuine parvoviral late promoter P38, whose activity is upregulated by NS1. Production of recPV takes place in producer cell lines upon co-transfection of the recombinant viral genome (containing the transgene) with a second plasmid harbouring the VP gene unit (Figure 5). The latter provides in trans the VP proteins needed for virus assembly, thus compensating for the disruption of the structural genes in the recombinant viral genome. The recombinant parvoviral particles generated in this way are DNA replication competent but propagation-defective and achieve transgene expression only in primarily infected cells as a one-hit event. The deletion of a portion of the structural genes gives the opportunity to insert transgenes up to approximately 1200 nt, while the insertion of larger transgenes strongly impairs virus production. Recombinant rodent parvoviruses have been constructed using MVM and H-1PV infectious plasmids as backbones. Examples of transgene products expressed by means of recombinant H-1PV vectors include pro-apoptotic apoptin [99], immunostimulatory cytokines/chemokines (e.g., human *IL2*, *CCL7*, and *CCL2*) [100], and antiangiogenic modulators (e.g., CXCL10 and CXCL4L1) [101,102]. Some of these recPVs are therapeutically promising, as they proved to have an enhanced anticancer activity in preclinical animal models. However, efficient production of these recombinants remains a major obstacle in their way to the clinic [98]. Co-transfection of the above helper system with a plasmid containing the adenoviral E2a, E4(orf6), and the VA RNA genes (e.g., pXX6 plasmid) improved the production of recPVs by more than 10-fold [91]. Based on these results, an Ad harbouring the VP gene unit was constructed and used as a helper. This VP expressing Ad further improved the recPV yields because it allowed cell lines that were difficult to transfect but efficient at producing recPVs (e.g., NB324K) to be used according to a protocol that relied entirely on virus infection [103]. Further research and development activities are worth conducting to optimize and scale-up recPV production.

## 6. Where Next for H-1PV?

Research on H-1PV as an oncolytic agent goes back to the 1960s. Since then, many efforts have been devoted to elucidating the virus life cycle, its strict interactions with the cancer cell host, and its anticancer properties. After more than 50 years of preclinical research, these efforts culminated with the aforementioned launch of the first clinical study in patients with recurrent glioblastoma. The results of the clinical study showed that H-1PV treatment is safe, well tolerated, and associated with first signs of anticancer efficacy. These results provide new impetus for novel research and development activities aimed at further strengthening the H-1PV anticancer profile and bringing into the clinic more potent H-1PV-based therapies that could improve clinical outcome.

In Section 5, we provided examples of how this could be achieved through the rational design of H-1PV-based combination therapies and/or the development of novel, more potent, second-generation H-1PV vectors. These studies are particularly important and should guarantee a portfolio of novel, more efficient PV-based treatments to be tested in clinical trials. All these strategies should aim to increase oncolysis not only in quantitative but also in qualitative terms. Indeed, how a certain treatment kills cancer cells determines the success of the therapy, as the treatments vary in their immunogenicity and, thus, engage the immune system to act against the cancer at different degrees. To maximize the role of the immune system in destroying cancer cells, combinations of H-1PV with other forms of immunotherapy are particularly promising strategies. This was observed within the frame of a compassionate use program in patients with glioblastoma (see Section 5).

To increase the success rate of the virus treatment, it is also very important to identify biomarkers that could predict the outcome of the therapy. Currently, we are still missing basic knowledge on the determinants that make a tumor susceptible or resistant to H-1PV infection. For instance, it remains elusive why some cancer cells sustain virus replication more efficiently than others. Also, the entry pathways used by H-1PV and the mechanisms underlying virus trafficking into the nucleus are largely uncharacterized. Studies in these areas may reveal key cellular modulators (either activators or repressors) of the virus life cycle that could help us to predict whether a certain patient is likely to benefit from virus treatment. This information may be used as part of a more personalized virus treatment in which a certain therapy is selected according to patient tumor genetic makeup. At the same time, these studies may provide the key to improving virus treatment, for instance by guiding us in the identification of new drugs that could reinforce virus replication and oncolysis in tumor cells or of new means to improve next-generation PV vectors (e.g., shRNA targeting negative modulators of the virus life cycle). These new developments may overcome current molecular restrictions that limit efficacy and thereby extend the success of H-1PV-based therapies.

## Figures and Tables

**Figure 1 viruses-11-00562-f001:**
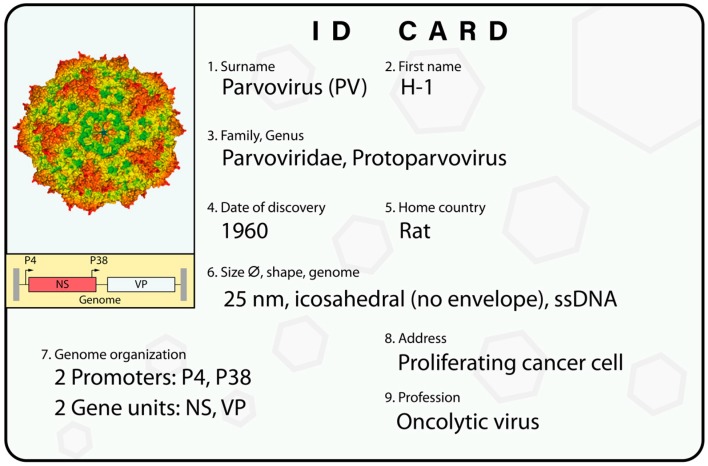
H-1PV’s ID card. An overview of H-1PV classification and main features. The virus genome is a single-stranded DNA (ssDNA) molecule including two promoters. The P4 promoter controls the nonstructural unit (NS), which encodes the NS1 and NS2 nonstructural proteins; and the P38 promoter regulates the expression of the VP gene unit, which encodes the VP1 and VP2 capsid proteins. At its extremities, the viral genome contains palindromic sequences (depicted in grey) that are important for virus DNA amplification. An in silico model of the virus capsid is shown [41]. See text for a more detailed description.

**Figure 2 viruses-11-00562-f002:**
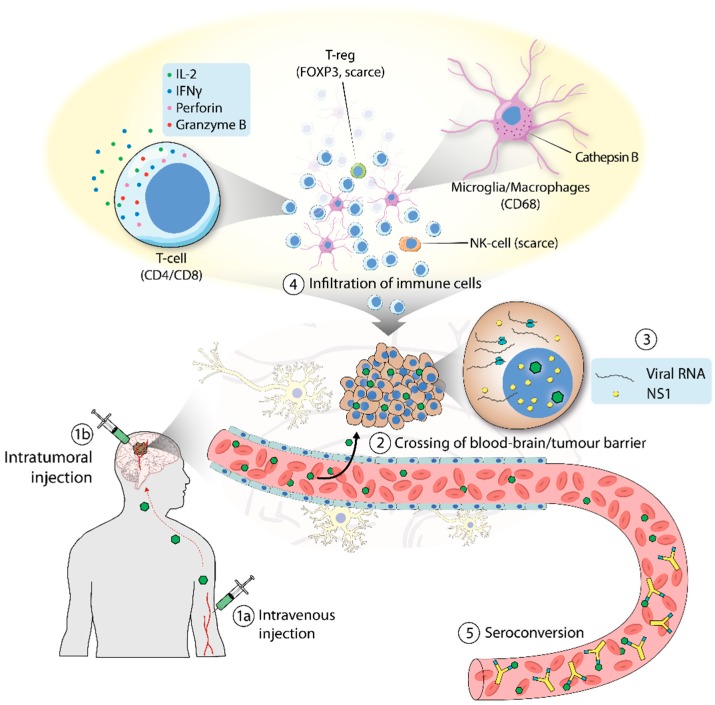
Clinical trial of H-1PV in patients with glioblastoma. (1) H-1PV was administered intratumorally (a) or intravenously (b). (2) H-1PV injected intravenously reached the brain tumor by crossing the blood–brain barrier. (3) H-1PV successfully infected cancer cells, which were positive for viral RNA and NS1 protein (although NS1 expression was below detection limits when the virus was given intravenously). (4) H-1PV induced immunoconversion of the tumor microenvironment (TME), which was characterized by infiltration of CD4+ and CD8+ T cells. T cells were found in their active state, as deduced from the expression of perforin and granzyme B. Microglia/microphages were also observed in the TME. These cells expressed high levels of cathepsin B. By contrast, only a small number of regulatory T cells (T-reg), which tested positive for FOXP3, and few natural killer (NK) cells were detected. (5) Seroconversion occurred after a few days, with the production of virus-neutralizing antibodies.

**Figure 3 viruses-11-00562-f003:**
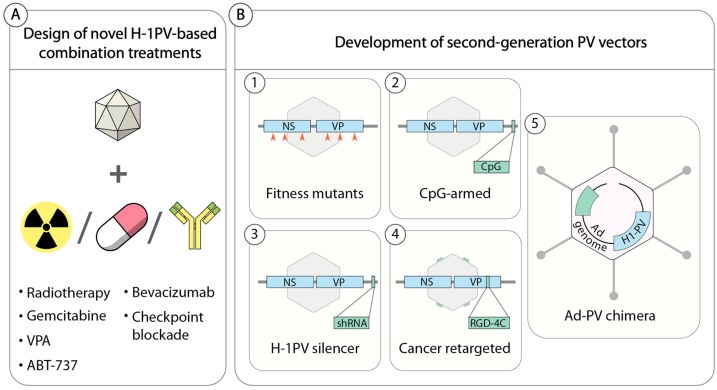
Improving H-1PV-based therapies. (**A**) Combination with drugs. H-1PV anticancer activity can be improved by combining the virus with other anticancer modalities. (**B**) Development of second-generation PV vectors. (1) H-1PV was successfully genetically modified by directed molecular evolution. By serially passaging H-1PV in semipermissive cancer cells, the virus acquired random mutations (orange arrowheads) that improved virus replication and spreading. (2 and 3) Functional elements (CpG motifs or an shRNA expression cassette) were inserted into the H-1PV genome without affecting its replication ability. These elements enhanced virus-mediated immune modulation and oncolysis, respectively. (4) Virus retargeting. The viral capsid was genetically modified by inserting an arginine-glycine-aspartic acid (RGD-4C) peptide, which improved cancer specificity at the level of virus cell entry. (5) Construction of Ad–PV chimeras. An engineered version of the H-1PV genome was inserted into a nonreplicative adenovirus (Ad) genome. The Ad–PV chimera brought the H-1PV genome into cancer cells and produced fully infectious H-1PV particles. The anticancer properties of Ad–PV chimera can be increased by inserting a therapeutic transgene into the Ad component of the Ad–PV hybrid genome (in green).

**Figure 4 viruses-11-00562-f004:**
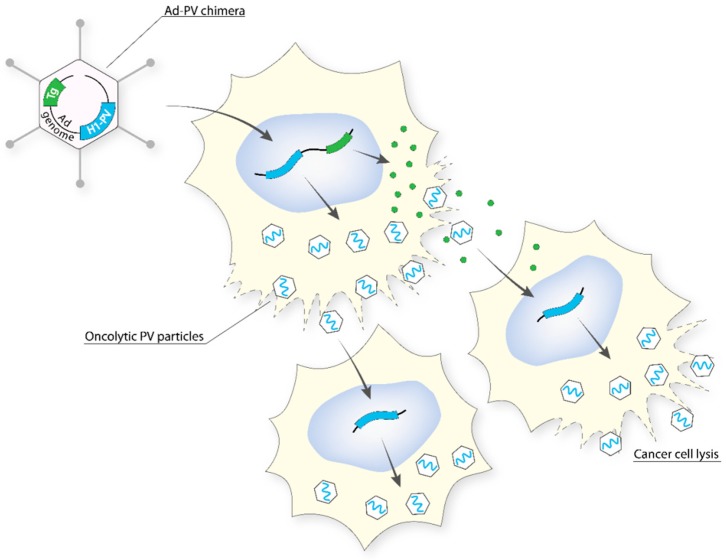
The Ad–PV chimera strategy. The entire oncolytic H-1PV genome (light blue) was inserted into a replication-defective E1 and E3-deleted Ad5 vector genome (Ad, in black). The chimera effectively delivers the PV genome into cancer cells, from which progeny PV particles are generated. Additionally, a transgene (in green) is expressed from the vector genome and can act intracellularly or extracellularly. These PV particles can infect neighbouring cancer cells, kill them, and induce secondary rounds of lytic infection, thus amplifying the initial cytotoxic activity of the chimera [97].

**Figure 5 viruses-11-00562-f005:**
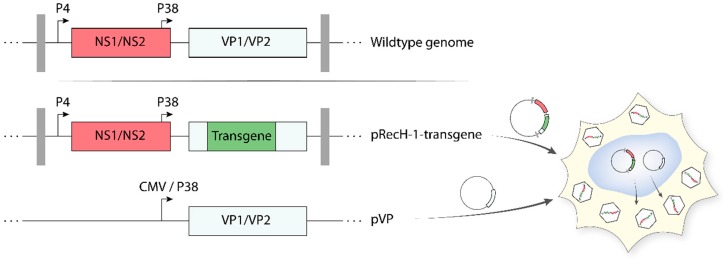
Production of recombinant H-1PV. For the production of recombinant H-1PV, a part of the VP1/VP2 region of the wild-type genome is removed and replaced with a transgene (in green). The VP gene unit under the control of a cytomegalovirus (CMV) or P38 promoter is provided in trans by a helper plasmid (pVP). Upon co-transfection of the plasmids, the producer cell line generates fully infectious, yet propagation-deficient, parvovirus particles containing the recombinant viral genome.

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
