# Peer review of "H-1 Parvovirus as a Cancer-Killing Agent: Past, Present, and Future"

_viruses, 2019, doi:10.3390/v11060562_

Reviewer 1 Report

This manuscript by Bretscher and Marchini represents an extensive, well-written and timely review of H-1PV as an anti-cancer agent.  It is not, however, entirely accurate in its coverage of the past use of this virus as an oncolytic agent.

Considering their own current interest, it is perhaps understandable for the authors to have confined themselves to H-1PV.  Unfortunately, however, this focus ignores parallel studies using other rodent protoparvoviruses as anti-cancer vectors.  The authors also only consider the use of H-1PV as a "live" virus, i.e. one capable of gene expression and DNA replication within human cancer cells, and the production of progeny virions allowing subsequent infection of additional cells within the tumor.  Indeed, this property is explicitly cited as a major goal for improvement on page 8, line 300.  This priority is rather at odds with the general movement in the field away from the idea of oncolytic viruses being a self-amplifying tumor toxic drug to emphasize more the induction of anti-tumor immune responses.  These responses do not necessarily require multiple rounds of infection, as discussed at length in the article in this volume by Angelova and Rommelaere, cited as ref 51. 

While the authors of the present manuscript do discuss various ways in which H-1PV has been modified to enhance its immunogenic potential, it is always in the context of propagation-competent virus.  Although much of the work with non-propagative, capsid-replacement vectors has been done with rodent protoparvoviruses other than H-1PV, it is not clear why the authors chose to ignore reports of such vectors derived from H-1PV, designed to express MCP-1, IL-2 or MCP-3/CCL7, for instance, published by the Rommelaere group.

Given that "past" in the title of this otherwise comprehensive review implies a historical treatment, one final concern is the statement that research on H-1PV as an oncolytic virus began in the 1980's (page 13, line 516). This timeline ignores the essential contributions of Helene Toolan, who not only discovered H-1PV in the late 1950s (1), but was the first to realize that its isolation from transplantable human tumors was not causative of the tumor, but opportunistic - thus that it displayed a tropism for human cancer cells (2).  In the 1960s, Toolan further showed that H-1PV suppressed virally- and chemically-induced tumors, as well as lowering the rate of spontaneous tumors in animal models (3-5).  Realizing that its ability to infect human tumor cells might be used therapeutically, Toolan actually performed the 'first in human' clinical trial (6).  Although this did not alter the course of the patients' cancers, the exercise was critical to the subsequent acceptance that administering H-1PV to cancer patients would be a low risk procedure.  If the authors of this review wish to put the use of H-1PV as an oncolytic agent in an accurate historical context, as well as a current and future perspective, I strongly suggest that Toolan's seminal contributions be fully acknowledged.

Specific Comments

Page 1, line 2:  since "PV" stands for parvovirus, this would be better as either "H-1PV as a ..." or "H-1 parvovirus as a ...".

Page 2, line 84:  "contains a linear single-stranded DNA molecule"

Page 4, line 137:  some may consider that dressing H-1 in a shirt and tie detracts from the otherwise serious nature of the review!

Page 4, line 146:  it would be worth mentioning here that, although sero-epidemiological studies indicate that H-1 is a rat virus, the laboratory strain used since its original isolation by Toolan was derived from a presumably adventitious infection of the human Hep-1 hepatoma cell line, transplanted in cortisone-immunosuppressed rats.  The current H-1PV therefore may have been selected for growth in human tumor cells, and may differ in this respect from authentic field isolates.

Page 6, line 252 on:  Dempe et al., (Int J Cancer 126, 2914-27, 2009) showed that the sensitivity of PDAC cell lines to H-1PV depends upon their expression of SMAD4.  The authors need to discuss this potential limitation of H-1PV in the treatment of pancreatic cancer.

Page 10, line 400 on:  it is not clear what the authors consider as "first generation vectors" for which these "second generation parvoviruses" represent an improvement.  Do you mean unmodified live virus or the capsid-replacement vectors pioneered by Rommelaere and colleagues - see general comments.

Page 11, line 426:  "subsequent" would be a more appropriate word than "successive" here.

Page 12, line 478:  "Allaume et al. (31) showed ...." - please insert reference here.

Page 13, line 516:  the statement here not strictly accurate - research on H-1PV as an oncolytic agent goes back to the pioneering studies of Toolan and colleagues in the 1960s - see general comments.

Useful historical references:

1) Toolan HW, Dalldore G, Barclay M, Chandra S & Moore AE.  An unidentified, filtrable agent isolated from transplanted human tumors. Proc Natl Acad Sci USA. 46(9):1256-8, 1960.

2) Toolan HW. A virus associated with transplantable human tumors. Bull N Y Acad Med. 37:305-10, 1961.

3) Toolan HW. Lack of oncogenic effect of the H-viruses for hamsters. Nature 214(5092):1036, 1967.

4) Toolan HW & Ledinko N. Inhibition by H-1 virus of the incidence of tumors produced by adenovirus 12 in hamsters. Virology. Jul;35(3):475-8, 1968.

5) Toolan HW, Rhode SL 3rd & Gierthy JF. Inhibition of 7,12-dimethylbenz(A)anthracene-induced tumors in Syrian hamsters by prior infection with H-1 parvovirus.  Cancer Res. 42(7):2552-5, 1982.

6) Toolan HW, Saunders EL, Southam CM, Moore AE & Levin AG. H-1 virus viremia in the human. Proc Soc Exp Biol Med. 19:711-5, 1965

Author Response

Dear Editor,

We are pleased to submit a new revised version of our manuscript  “H-1 parvovirus as a cancer-killing agent: past, present and future” by Bretscher and Marchini for publication in Viruses.

We were glad to know that the referees found our review interesting. We greatly appreciated reviewers’ efforts to carefully read the manuscript and we were grateful to receive valuable advices and constructive criticism which improved our review. Please find below a point-by point response to reviewers’ comments. Four your easy please also find another version of the manuscript where we highlight all changes in yellow.

We thank you and the reviewers for your help.

We hope that you will find our revised manuscript suitable for publication in your Journal.

Kind regards,

Antonio Marchini

Response to reviewers

Reviewer 1

This manuscript by Bretscher and Marchini represents an extensive, well-written and timely review of H-1PV as an anti-cancer agent.  It is not, however, entirely accurate in its coverage of the past use of this virus as an oncolytic agent. 

and

Given that "past" in the title of this otherwise comprehensive review implies a historical treatment, one final concern is the statement that research on H-1PV as an oncolytic virus began in the 1980's (page 13, line 516). This timeline ignores the essential contributions of Helene Toolan, who not only discovered H-1PV in the late 1950s (1), but was the first to realize that its isolation from transplantable human tumors was not causative of the tumor, but opportunistic - thus that it displayed a tropism for human cancer cells (2).  In the 1960s, Toolan further showed that H-1PV suppressed virally- and chemically-induced tumors, as well as lowering the rate of spontaneous tumors in animal models (3-5).  Realizing that its ability to infect human tumor cells might be used therapeutically, Toolan actually performed the 'first in human' clinical trial (6).  Although this did not alter the course of the patients' cancers, the exercise was critical to the subsequent acceptance that administering H-1PV to cancer patients would be a low risk procedure.  If the authors of this review wish to put the use of H-1PV as an oncolytic agent in an accurate historical context, as well as a current and future perspective, I strongly suggest that Toolan's seminal contributions be fully acknowledged.

AM: Thank you very much for the constructive criticism and for the valuable information provided. This has been enclosed in the new version of the manuscripts at page 2 lines 82-87.

Considering their own current interest, it is perhaps understandable for the authors to have confined themselves to H-1PV.  Unfortunately, however, this focus ignores parallel studies using other rodent protoparvoviruses as anti-cancer vectors. 

AM: While the main focus of the review remains on H-1PV and propagation competent H-1PV based derivatives, new references has been enclosed in the new version of the manuscript which acknowledge the contribution of colleagues working on other rodent protoparvoviruses (see page 2 lines 75-79 and new references 19-22).

The authors also only consider the use of H-1PV as a "live" virus, i.e. one capable of gene expression and DNA replication within human cancer cells, and the production of progeny virions allowing subsequent infection of additional cells within the tumor.  Indeed, this property is explicitly cited as a major goal for improvement on page 8, line 300.  This priority is rather at odds with the general movement in the field away from the idea of oncolytic viruses being a self-amplifying tumor toxic drug to emphasize more the induction of anti-tumor immune responses.  These responses do not necessarily require multiple rounds of infection, as discussed at length in the article in this volume by Angelova and Rommelaere, cited as ref 51. While the authors of the present manuscript do discuss various ways in which H-1PV has been modified to enhance its immunogenic potential, it is always in the context of propagation-competent virus.  Although much of the work with non-propagative, capsid-replacement vectors has been done with rodent protoparvoviruses other than H-1PV, it is not clear why the authors chose to ignore reports of such vectors derived from H-1PV, designed to express MCP-1, IL-2 or MCP-3/CCL7, for instance, published by the Rommelaere group.

AM: As suggested by the reviewer, in the new version of the manuscript we have inserted a new paragraph describing the recPV vectors (page 14 and 15  lines 576-612 and new figure 5). We have also explained why in our opinion it is important to devote efforts in improving H-1PV replication in cancer cells (lines 338-347).

Specific Comments

 Page 1, line 2:  since "PV" stands for parvovirus, this would be better as either "H-1PV as a ..." or "H-1 parvovirus as a ...".

AM:  Title has been corrected..

Page 2, line 84:  "contains a linear single-stranded DNA molecule".  AM: Added, Thank you

Page 4, line 137:  some may consider that dressing H-1 in a shirt and tie detracts from the otherwise serious nature of the review!

AM: We have revised figure1 accordingly.

Page 4, line 146:  it would be worth mentioning here that, although sero-epidemiological studies indicate that H-1 is a rat virus, the laboratory strain used since its original isolation by Toolan was derived from a presumably adventitious infection of the human Hep-1 hepatoma cell line, transplanted in cortisone-immunosuppressed rats.  The current H-1PV therefore may have been selected for growth in human tumor cells, and may differ in this respect from authentic field isolates.

AM: Thank you very much for this helpful suggestion. The information has been enclosed in the new version of the manuscript at lines 97-99.

Page 6, line 252 on:  Dempe et al., (Int J Cancer 126, 2914-27, 2009) showed that the sensitivity of PDAC cell lines to H-1PV depends upon their expression of SMAD4.  The authors need to discuss this potential limitation of H-1PV in the treatment of pancreatic cancer.

AM: Thank you for the valuable comment. The correction has been done accordingly (lines 283-286)

Page 10, line 400 on:  it is not clear what the authors consider as "first generation vectors" for which these "second generation parvoviruses" represent an improvement.  Do you mean unmodified live virus or the capsid-replacement vectors pioneered by Rommelaere and colleagues - see general comments.

AM: We clarified this point giving a  new definition and distinguishing propagation competent and propagation defective H-1PV vectors in the new version of the manuscript (lines 457 and 576)

Page 11, line 426:  "subsequent" would be a more appropriate word than "successive" here.

AM: Corrected. Thank you.

Page 12, line 478:  "Allaume et al. (31) showed ...." - please insert reference here. AM: Reference inserted

Page 13, line 516:  the statement here not strictly accurate - research on H-1PV as an oncolytic agent goes back to the pioneering studies of Toolan and colleagues in the 1960s - see general comments.

AM: Done. Thank you very much.  

Useful historical references:

 1) Toolan HW, Dalldore G, Barclay M, Chandra S & Moore AE.  An unidentified, filtrable agent isolated from transplanted human tumors. Proc Natl AcadSci USA. 46(9):1256-8, 1960. 

2) Toolan HW. A virus associated with transplantable human tumors. Bull N Y Acad Med. 37:305-10, 1961.

3) Toolan HW. Lack of oncogenic effect of the H-viruses for hamsters. Nature 214(5092):1036, 1967.

4) Toolan HW &Ledinko N. Inhibition by H-1 virus of the incidence of tumors produced by adenovirus 12 in hamsters. Virology. Jul;35(3):475-8, 1968.

5) Toolan HW, Rhode SL 3rd &Gierthy JF. Inhibition of 7,12-dimethylbenz(A)anthracene-induced tumors in Syrian hamsters by prior infection with H-1 parvovirus.  Cancer Res. 42(7):2552-5, 1982.

6) Toolan HW, Saunders EL, Southam CM, Moore AE & Levin AG. H-1 virus viremia in the human. Proc SocExpBiol Med. 19:711-5, 1965

AM: All the historical references have been inserted.  Thank you!

Reviewer 2

Review of Manuscript “H-1PV parvovirus as a cancer-killing agent: past present and future” by Bretscher et al..

The authors present a very comprehensive review of rat parvovirus H-1 as a promising agent for tumor therapy through selective replication of the virus in cancer cells leading to killing and eradication of the later. Whereas the sections describing the general properties of H-1 have already been reviewed in a number of other publications and may therefore be slightly shortened, the sections describing the clinical studies performed with H-1 and novel strategies to improve the clinical outcome are highly informative. Especially the later sections, addressing both possible modifications of the virus and combinations with other kinds of treatments such as radiotherapy or chemical anticancer modalities, are of outstanding interest for researchers working in the field of oncolytic viruses and readers looking for a good overview of these novel developments. The figures are very informative and nicely summarize the descriptions of the text.

AM: We were happy to know that the reviewer found interesting our review.  Thank you very much for the positive feedback.

Reviewer 2 Report

Review of Manuscript “H-1PV parvovirus as a cancer-killing agent: past present and future” by Bretscher et al..

The authors present a very comprehensive review of rat parvovirus H-1 as a promising agent for tumor therapy through selective replication of the virus in cancer cells leading to killing and eradication of the later. Whereas the sections describing the general properties of H-1 have already been reviewed in a number of other publications and may therefore be slightly shortened, the sections describing the clinical studies performed with H-1 and novel strategies to improve the clinical outcome are highly informative. Especially the later sections, addressing both possible modifications of the virus and combinations with other kinds of treatments such as radiotherapy or chemical anticancer modalities, are of outstanding interest for researchers working in the field of oncolytic viruses and readers looking for a good overview of these novel developments. The figures are very informative and nicely summarize the descriptions of the text. 

Author Response

(The authors gave the same response as above.)
